# Impact of initial vancomycin pharmacokinetic/pharmacodynamic parameters on the clinical and microbiological outcomes of methicillin-resistant *Staphylococcus aureus* bacteremia in children

**Reenar Yoo** [1], **Hyejin So**[1], **Euri Seo**[2], **Mina Kim**[3], **Jina Lee**[1]*

1 Department of Pediatrics, Asan Medical Center, University of Ulsan College of Medicine, Seoul, South Korea, 2 Department of Pediatrics, Dongguk University Ilsan Hospital, Dongguk University College of Medicine, Goyang, Republic of Korea, 3 Department of Laboratory Medicine, Asan Medical Center, University of Ulsan College of Medicine, Seoul, South Korea

* entier@amc.seoul.kr

**Data Availability Statement:** All relevant data are within the manuscript and its Supporting information files.

## Abstract

Optimal vancomycin exposure is important to minimize treatment failure of methicillin-resistant *Staphylococcus aureus* (MRSA) bacteremia. We aimed to analyze the impact of initial vancomycin pharmacokinetic/pharmacodynamic (PK/PD) parameters, including the initial vancomycin $C_{trough}$ and the area under the curve (AUC)/minimal inhibitory concentration (MIC) on the outcomes of pediatric MRSA bacteremia. The study population consisted of hospitalized children aged between 2 months and 18 years with MRSA bacteremia, in whom $C_{trough}$ was measured at least one time within the time period of January 2010 to March 2018. Demographic profiles, underlying diseases, and clinical/microbiological outcomes were abstracted retrospectively. During the study period, 73 cases of MRSA bacteremia occurred in children with a median age of 12.4 months. Severe clinical outcomes leading to intensive care unit stay and/or use of mechanical ventilation occurred in 47.5% (35/73); all-cause 30-day mortality was 9.7% (7/72). The median dosage of vancomycin was 40.0 mg/kg/day. There was a weak linear relationship between $C_{trough}$ and the corresponding AUC/MIC ($r = 0.235$). ROC curves for achieving an AUC/MIC of 300 suggested that the initial $C_{trough}$ at 10 μg/mL could be used as a cut-off value with a sensitivity of 90.5% and a specificity of 44%. Although persistent bacteremia at 48–72 hours after vancomycin administration was observed more frequently when the initial $C_{trough}$ was < 10 μg/mL and initial AUC/MIC was < 300, initial AUC/MIC < 300 was the only risk factor associated with persistent bacteremia at 48–72 hours (adjusted OR 3.05; 95% CI, 1.07–8.68). Initial $C_{trough}$ and AUC/MIC were not associated with 30-day mortality. Although there was a weak relationship between $C_{trough}$ and AUC/MIC, initial AUC/MIC < 300 could be used as a predictor of persistent MRSA bacteremia at 48–72 hours. Further prospective data on optimal vancomycin dosing are necessary to improve clinical and microbiological outcomes in pediatric MRSA bacteremia.

**Funding:** The authors received no specific funding for this work.

**Competing interests:** The authors have declared that no competing interests exist.

## Introduction

Methicillin-resistant *Staphylococcus aureus* (MRSA) is one of the major health concerns in both healthcare-associated and community-acquired infections in children, and clinical outcomes of MRSA bacteremia are considered to be worse than cases caused by methicillin-susceptible *S. aureus* (MSSA) [1, 2]. Vancomycin is an important treatment option for invasive MRSA infections [1]. To achieve favorable clinical outcomes while avoiding vancomycin toxicity associated with an overdose, the methods of monitoring of vancomycin administration have been studied. The results of these studies suggested that a ratio of the area under the curve to the minimum inhibitory concentration (AUC/MIC) has been correlated with efficacy in experiments conducted in vitro [3]. Although there is an assumption that a vancomycin $C_{trough}$ of $> 15$ μg/mL when the MIC is $\leq 1$ μg/mL can be used as a good surrogate marker for AUC/MIC ratio of $\geq 400$ with the MIC determined by broth microdilution (BMD) for achieving clinical effectiveness in adults [4], vancomycin dosing information to ensure optimal drug exposure in the pediatric population remains limited [5]. However, recent studies have suggested that high concentrations of vancomycin may not be necessary for pediatric populations; vancomycin $C_{trough}$ 7–10 μg/mL could predict the achievement of AUC/MIC $> 400$ at a dose of 15 mg/kg/dose every 6 hours in the case of patients with normal renal function, if the MIC $\leq 1$ μg/mL [6], and the risk of vancomycin nephrotoxicity may increase if $C_{trough}$ exceeds the predetermined range [7–9]. In addition, $C_{trough}$ is not highly correlated with AUC/MIC and is insufficient as a surrogate marker for clinical outcomes, not only in pediatric populations but also in adults [10, 11].

According to our previous study, initial vancomycin $C_{trough}$ of $< 10$ μg/mL was associated with microbiologic failure at 48 hours after vancomycin administration for pediatric MRSA bacteremia [12]. Another study on the impact of vancomycin $C_{trough}$ on the duration of MRSA bacteremia in children also suggested a $C_{trough}$ of $> 10$ μg/mL within the first 72 hours was the most predictive factor for persistent MRSA bacteremia lasting for $> 3$ days [13]. However, both studies focused only on the parameter of $C_{trough}$ without any evaluation of the impact of AUC/MIC on the clinical and microbiological outcomes.

Herein, we conducted a study elucidating the pharmacokinetics/pharmacodynamics (PK/PD) parameters, including both initial $C_{trough}$ and AUC/MIC, for optimal vancomycin dosing to achieve favorable clinical and microbiological outcomes in pediatric MRSA bacteremia. In addition, we aimed to determine whether $C_{trough}$ could be used as an optimal surrogate marker replacing AUC/MIC in clinical practice.

## Materials and methods

### Study design

This retrospective study was performed at a single center, the Asan Medical Center Children's Hospital, which is a 253-bed academic-tertiary medical center located in Seoul, South Korea. Pediatric patients aged between 2 months and 18 years who were admitted during the study period from January 2010 to March 2018 with culture-confirmed MRSA bacteremia were eligible for inclusion. Patients were included if they received vancomycin for at least 48 hours, and their initial $C_{trough}$ had been sampled before the 4th or 5th dose within the 30 minutes before the next dose of vancomycin. The following MRSA bacteremia cases were excluded from this study: patients with chronic renal diseases requiring renal replacement therapy such as hemodialysis or peritoneal dialysis, and cases treated in the neonatal intensive care unit.

The analysis only included the first MRSA isolates detected during a single clinical episode occurring within a 4-week period, and duplicates from the same patient were excluded.

Demographic profile and clinical data including primary sites of infection, underlying diseases, duration of hospital stay, and blood laboratory results, including serum creatinine, and initial vancomycin dose were extracted from the electronic medical records. The antimicrobial susceptibilities of the *S. aureus* isolates were decided using a MicroScan Walk-Away 96-Combo Pos 28 panel (Siemens, West Sacramento, CA, USA).

The primary outcome of this study is to clarify the relationship between the initial $C_{trough}$ and AUC/MIC, and the potential impact of the variables on clinical and microbiological outcomes. This study was approved by the Institutional Review Board of Asan Medical Center with a waiver of informed consent due to the study being retrospective and using de-identified data collection and analysis (IRB No. 2019–1638).

## Definitions

Most of the definitions used in this study, including fever, the primary focus of infection, and persistent bacteremia, were the same as those described in a previous study [12]. Recurrent MRSA bacteremia was defined as another MRSA bacteremia event 1 month after MRSA bacteremia resolution. Co-infection was defined as the case in which clinically significant bacteria were identified from the blood culture at the same time as the diagnosis of MRSA bacteremia. Clinically severe cases included those who required an intensive care unit stay, mechanical ventilation, and/or fatal cases. Acute kidney injury (AKI) was defined as an increase in serum creatinine levels by 0.5 mg/dL [14]. The clinically significant renal toxicity of vancomycin was defined as an additional need for renal replacement therapy during vancomycin treatment or the need to replace vancomycin with an alternate antibiotic because of serum creatinine elevation.

## Clinical and microbiological outcomes

The clinical and microbiological outcomes were analyzed according to the initial vancomycin $C_{trough}$, vancomycin MIC of MRSA isolates, and initial AUC/MIC, respectively. As measures of clinical outcomes, recurrence of MRSA bacteremia and 30-day all-cause mortality were evaluated. Microbiological outcomes were determined using the time-to-negative conversion of blood culture and failure of clearance of bacteremia at 48–72 hours after the initiation of vancomycin administration.

## Pharmacokinetic data

PK parameters including AUC and vancomycin clearance were calculated as follows:

- AUC = vancomycin daily dose (mg)/vancomycin clearance (mL/hr/kg) [15].

- Vancomycin clearance (mL/hr/kg) = $0.248 \times Wt (Kg)^{0.75} \times [0.48/serum\ Cr\ (mg/dL)]^{0.361} \times [\ln(age\ in\ days)/7.8]^{0.995}$ [5].

## Statistical analysis

Categorical data were analyzed with $X^2$ tests and Fisher's exact test, and continuous variables with independent t-tests or one-way ANOVA. Spearman's correlation coefficient was used to evaluate relationships between $C_{trough}$ and AUC/MIC. A receiver operating characteristic (ROC) curve was used to determine the sensitivity and specificity of each PK/PD parameter. A multivariate logistic regression analysis was used to assess the potential impact of the variables on clinical and microbiological outcomes. A $P$ value of $< 0.05$ was considered statistically

significant for the comparisons. Statistical Program for Social Science release 21 (SPSS Inc., Chicago, IL, USA) was used for all statistical calculations.

## Results

### Patient characteristics

During the study period from January 2010 to March 2018, a total of 98 patients aged between 2 months and 18 years experienced MRSA bacteremia in our institute. Excluding 25 patients because of hemodialysis (n = 17), unavailable data of $C_{trough}$ (n = 3), and use of an alternative antibiotic instead of vancomycin (n = 5), a total of 73 patients who fulfilled the inclusion criteria were analyzed in this study. The patients' median age was 12.4 months (range, 2 months–17.3 years), and 97.3% (71 out of 73) had underlying medical comorbidities, of which congenital heart disease was the most common underlying disease (28.8%; 21/73) (Table 1). With the exception of one case, all other cases of MRSA bacteremia were healthcare-associated infections. Bacterial co-infection occurred in a total of 5 patients, all of which were bacteremia due to *Enterococcus faecalis* (n = 2), *Klebsiella aerogenes* (n = 1), *K. pneumoniae* (n = 1), and *Pseudomonas aeruginosa* (n = 1).

A primary focus of MRSA bacteremia was found in 57.5% (42 out of 73), which consisted of central vascular catheters (59.5%, 25 out of 42), pneumonia (19.0%, 8 out of 42), skin and soft tissue infections (16.7%, 7 out of 42), and ventriculoperitoneal shunt (4.8%, 2 out of 42); the remaining 42.5% (31 out of 73) were primary bacteremia. Among the 25 cases of central line-associated bloodstream infection, 12 (48%) retained the central vascular catheter as salvage therapy. The simultaneous use of other antibiotics with vancomycin was observed in 86.3% (63 out of 73).

Fever was the most common initial symptom (75.3%, 55 out of 73). One out of 73 patients with MRSA bacteremia was transferred to another hospital during treatment, allowing analysis of mortality and recurrence rates in a total of 72 patients: all-cause 30-day mortality was 9.7% (7 out of 72). Recurrent MRSA bacteremia and persistent bacteremia at 48–72 hours occurred in 19.4% (14/72) and 39.7% (29/73) patients, respectively. Although AKI occurred in 3 out of 73 in this study, there was no clinically significant renal toxicity associated with vancomycin use.

### Vancomycin MIC of MRSA isolates

All of the 73 MRSA isolates were susceptible to vancomycin with MIC $\leq$ 2.0 μg/mL. Most of the MRSA isolates belonged to the group with vancomycin MIC 1.0 μg/mL (n = 53; 72.6%), followed by the group with vancomycin MIC $\leq$ 0.5 μg/mL (n = 10; 13.7%), and 2.0 μg/mL (n = 10; 13.7%), respectively (Table 2). During the study period, the proportion of MRSA isolates with vancomycin MIC $\leq$ 0.5 μg/mL showed a decreasing trend; 30.8% (2010–2012), 3.3% (2013–2015), and 5.9% (2016–2018) (*P* for trend = 0.246).

### PK/PD of vancomycin use

The initial median dose of vancomycin was 40.0 mg/kg/day [interquartile range (IQR), 40.0–55.0 mg/kg/day], and an initial vancomycin dose of $\geq$ 60 mg/kg/day was used in 28.3% of cases (17 out of 73). Initial median $C_{trough}$ was 6.4 μg/mL (IQR, 4.3–11.6 μg/mL); 71.2% (52 out of 73) had an initial $C_{trough} \leq$ 10 μg/mL and an initial $C_{trough} >$ 15 μg/mL was achieved in only 8 patients. The median initial AUC/MIC was 336.4 (IQR, 271.4–422.6), and an initial AUC/MIC $\geq$ 300 and $\geq$ 400 was achieved in 64.4% and 32.9%, respectively. There was a weak positive linear relationship between $C_{trough}$ and AUC/MIC (*r* = 0.235) (Fig 1).

**Table 1. Demographic and clinical characteristics of 73 pediatric patients with MRSA bacteremia.**

| Characteristics | Number of cases (%) or median values (interquartile range) |
|---|---|
| Median age, months | 12.4 (5.3–36.1) |
| Sex (number; % of male) | 43 (58.9%) |
| Hospital stay before the onset of the MRSA bacteremia, days | 13.0 (4.0–33.0) |
| Presence of bacterial co-infection[a] | 5 (6.8%) |
| Presence of invasive device [b] | 58 (79.5%) |
| Presence of fever $\geq$ 38˚C at the onset of bacteremia | 55 (75.3%) |
| Presence of underlying diseases | 71 (97.3%) |
| Congenital heart disease | 21 (28.8%) |
| Malignancy | 12 (16.4%) |
| Chronic lung disease | 4 (5.5%) |
| Neurologic diseases | 5 (6.8%) |
| Others [c] | 15 (20.5%) |
| Primary focus of MRSA bacteremia | |
| None | 31 (42.5%) |
| Central vascular catheter | 25 (34.2%) |
| Lung | 8 (11.0%) |
| Others [d] | 9 (12.3%) |
| Initial laboratory findings | |
| Serum WBC (/uL) | 13,500 (8,600–19,300) |
| CRP (mg/dL) | 3.1 (0.8–6.7) |
| Serum Creatinine (mg/dL) | 0.3 (0.2–0.4) |
| Parameters related to vancomycin regimen | |
| Initial vancomycin dose (mg/kg/day) | 40.0 (40.0–55.0) |
| Initial vancomycin $C_{trough}$ concentration (mg/L) | 6.4 (4.3–11.6) |
| Initial AUC/MIC | 336.4 (271.4–422.6) |
| Simultaneous use of other antibiotics (%) | 63 (86.3%) |
| Recurrent bacteremia [e] | 14 (19.4%) |
| Persistent bacteremia at 48–72 hours | 29 (39.7%) |
| Acute kidney injury | 3 (4.1%) |
| All-cause 30 day-mortality [e] | 7 (9.7%) |

[a] Bacterial co-infection occurred in a total of 5 patients, all of which were bacteremia due to *Enterococcus faecalis* (n = 2), *Klebsiella aerogenes* (n = 1), *K. pneumoniae* (n = 1), and *Pseudomonas aeruginosa* (n = 1).

[b] Invasive devices included central vascular catheter (n = 46), gastrostomy (n = 5), tracheostomy (n = 5), ventriculo-peritoneal shunt (n = 2).

[c] Others included congenital diaphragmatic hernia (n = 2), congenital megacolon (n = 2), short bowel syndrome (n = 3), acute respiratory distress syndrome (n = 3), omphalocele (n = 1), trachea-esophagus fistula (n = 1), jejunal atresia (n = 1), pseudohypoaldosteronism (n = 1), VACTERL (vertebral defects, anal atresia, cardiac defects, tracheoesophageal fistula, renal anomalies, and limb abnormalities association) (n = 1).

[d] Others included skin and soft tissue infection (n = 7), ventriculo-peritoneal shunt infection (n = 2).

[e] One out of 73 patients with MRSA bacteremia was transferred to another hospital during treatment, allowing analysis of mortality and recurrence rates in a total of 72 patients excluding it.

ROC curves for predicting the achievement of AUC/MIC $\geq$ 300 suggested that the initial $C_{trough}$ of 10 μg/mL could be used as a cut-off value with a sensitivity of 90.5% and a specificity of 44.0%; an AUC of 0.721 [95% confidence interval (CI), 0.593–0.849]. An achievement of AUC/MIC $\geq$ 400 suggested sensitivity of 57.1% and specificity of 78.8% at the cut-off value of initial $C_{trough}$ of 10 μg/mL.

**Table 2. Annual trend of vancomycin MIC of MRSA isolates.**

| Year | Vancomycin MIC range determined by Microscan | | |
|---|---|---|---|
| | < 0.5 µg/mL | 1.0 µg/mL | 2.0 µg/mL |
| 2010–2012 | 30.8% (8/26) | 53.8% (14/26) | 15.4% (4/26) |
| 2013–2015 | 3.3% (1/30) | 86.7% (26/30) | 10.0% (3/30) |
| 2016–2018 | 5.9% (1/17) | 76.5% (13/17) | 17.6% (3/17) |

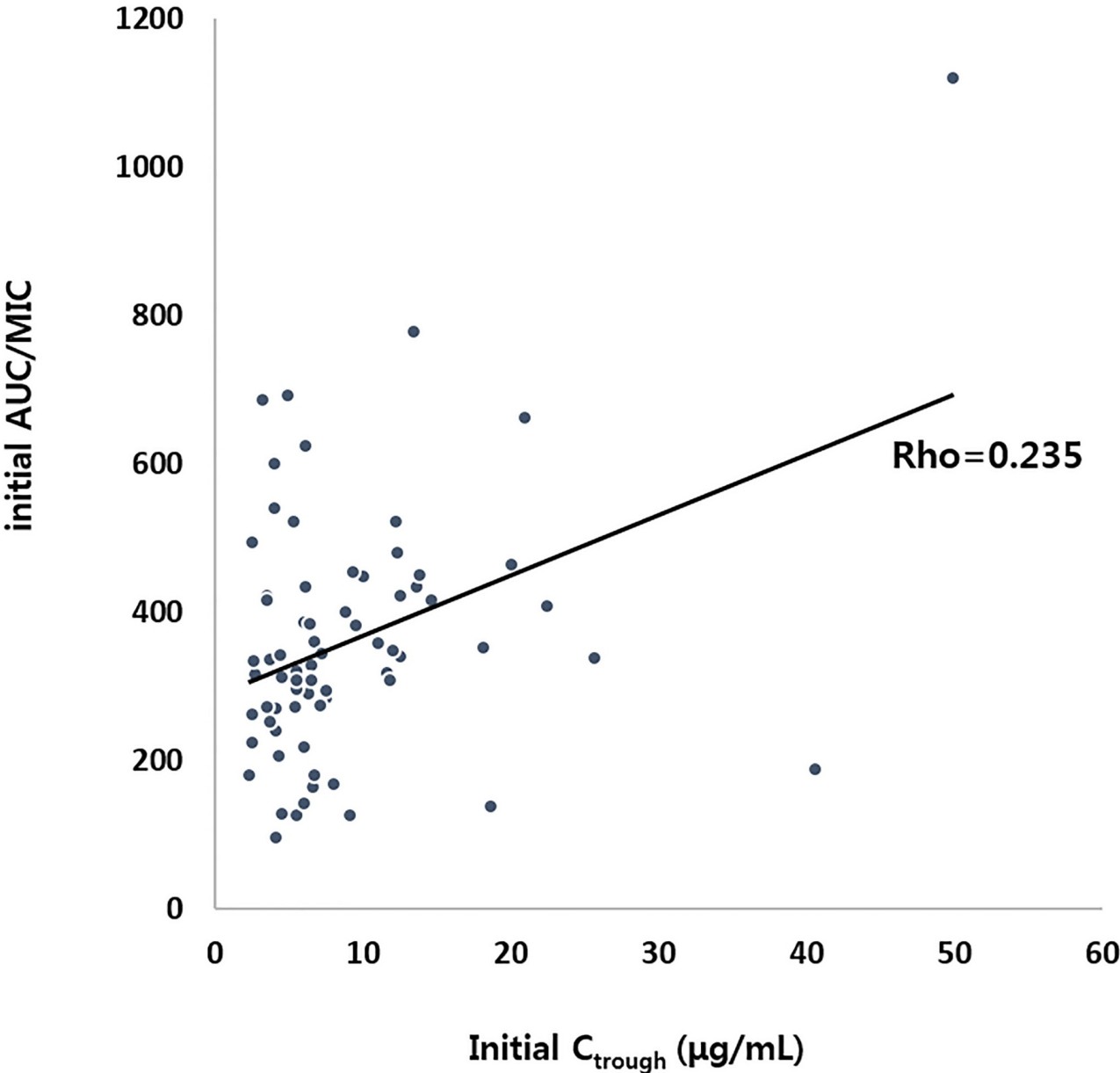

**Fig 1. Correlation between initial C trough and AUC/MIC.**

**Table 3. Clinical and microbiological outcomes of children with MRSA bacteremia according to the pharmacokinetic parameters of vancomycin and the MIC of MRSA isolates.**

| | Microbiological outcome | | Clinical outcome [a] | |
| --- | --- | --- | --- | --- |
| | Time to negative conversion (days) | Persistent bacteremia at 48–72 hours | Recurrent MRSA bacteremia [b] (n = 14) | All-cause 30-day mortality (n = 7) |
| Initial vancomycin $C_{trough}$ (µg/mL) | | | | |
| < 10 (n = 52) | 4.1 ± 4.0 | 23 (44.2%) | 9/52 (17.3%) | 6/52 (11.5%) |
| ≥ 10 (n = 21) | 2.5 ± 1.5 | 4 (19.0%) | 5/20 (25.0%) | 1/20 (5.0%) |
| $P$-value [c] | 0.018 | 0.044 | 0.460 | 0.402 |
| < 15 (n = 65) | 3.7 ± 3.7 | 24 (36.9%) | 12/64 (18.8%) | 7/64 (10.9%) |
| ≥ 15 (n = 8) | 2.9 ± 1.9 | 3 (37.5%) | 2/8 (25.0%) | 0/8 (0%) |
| $P$-value [c] | 0.546 | 0.975 | 0.674 | Not applicable |
| Initial vancomycin AUC/MIC | | | | |
| < 300 (n = 25) | 3.9 ± 3.0 | 14 (53.8%) | 6/25 (24.0%) | 3/25 (12.0%) |
| ≥ 300 (n = 48) | 3.4 ± 3.8 | 13 (27.7%) | 8/47 (17.0%) | 4/47 (8.5%) |
| $P$-value [c] | 0.632 | 0.026 | 0.476 | 0.634 |
| < 400 (n = 49) | 3.2 ± 2.6 | 19 (38.8%) | 10/48 (20.8%) | 4/48 (8.3%) |
| ≥ 400 (n = 24) | 4.4 ± 4.9 | 8 (33.3%) | 4/24 (16.7%) | 3/24 (12.5%) |
| $P$-value [c] | 0.184 | 0.651 | 0.674 | 0.574 |
| Vancomycin MIC (µg/mL) | | | | |
| ≤ 1.0 (n = 63) | 6.5 ± 17.4 | 21 (33.3%) | 10/62 (16.1%) | 7/62 (11.3%) |
| 2.0 (n = 10) | 4.0 ± 3.5 | 6 (60.0%) | 4/10 (40.0%) | 0/10 (0%) |
| $P$-value [c] | 0.659 | 0.105 | 0.077 | Not applicable |
| Initial vancomycin dose | | | | |
| ≥ 60 mg/kg/day (n = 17) | 8.2 ± 23.5 | 3 (17.6%) | 5/17 (29.4%) | 1/17 (5.9%) |
| < 60 mg/kg/day (n = 56) | 5.6 ± 13.3 | 24 (42.9%) | 9/55 (16.4%) | 6/55 (10.9%) |
| $P$-value [c] | 0.154 | 0.059 | 0.235 | 0.541 |

[a] One out of 73 patients with MRSA bacteremia was transferred to another hospital during treatment, allowing analysis of mortality and recurrence rates in a total of 72 patients excluding it.

[b] Recurrent MRSA bacteremia defined as another MRSA bacteremia event 1 month after MRSA bacteremia resolution.

[c] Categorical data were analyzed with $X^2$ tests.

## Impact on clinical and microbiological outcomes

Compared to the group with initial $C_{trough} \geq 10$ µg/mL, persistent bacteremia at 48–72 hours was observed more frequently in those with an initial $C_{trough} < 10$ µg/mL (44.2% vs 19.0%; $P = 0.044$) (Table 3). The mean time for negative conversion of MRSA bacteremia in those with an initial $C_{trough} < 10$ µg/mL was 4.1 ± 4.0 days, but it was 2.5 ± 1.5 days in those with a $C_{trough} \geq 10$ µg/mL ($P = 0.018$). However, there was no difference in the percentage of persistent bacteremia or duration of MRSA bacteremia with a cut off value of $C_{trough}$ of 15 µg/mL. In addition, there was no statistical significance between the initial vancomycin $C_{trough}$ and the 30-day mortality rate or recurrence of MRSA bacteremia.

Persistent bacteremia at 48–72 hours was observed more frequently in the group with initial AUC/MIC < 300 than those with AUC/MIC ≥ 300 (53.8% vs 27.7%; $P = 0.026$). However, in terms of initial microbiological outcomes, there was no statistically significant difference between the two groups with an initial AUC/MIC < 400 and ≥ 400. Furthermore, the time to negative conversion of MRSA bacteremia, 30-days mortality, or recurrent MRSA infection were indifferent to the initial AUC/MIC.

**Table 4. Risk factors for persistence of MRSA bacteremia at 48–72 hours and 30-day all-cause mortality.**

| Variable | Persistent bacteremia at 48–72 hours | | 30-day all-cause mortality | |
|---|---|---|---|---|
| | OR (95% CI) | | OR (95% CI) | |
| | Unadjusted | Adjusted | Unadjusted | Adjusted |
| Age | 1.00 (0.99–1.01) | 1.00 (0.99–1.01) | 1.01 (0.99–1.02) | 1.00 (0.99–1.02) |
| Bacterial co-infection [a] | 1.15 (0.18–7.34) | 1.37 (0.20–9.49) | 0.00 | 0.00 |
| Presence of invasive device [b] | 1.81 (0.51–6.37) | 1.60 (0.42–6.13) | 2.57 (0.30–22.07) | 2.42 (0.27–21.76) |
| Presence of primary focus | 2.38 (0.87–6.51) | 1.92 (0.66–5.56) | 1.87 (0.44–7.89) | 1.85 (0.41–8.28) |
| Initial $C_{trough}$ < 10 µg /mL | 3.37 (1.00–11.41) | 3.14 (0.86–11.41) | 1.73 (0.34–8.91) | 1.42 (0.26–7.78) |
| Initial AUC/MIC < 300 | 3.43 (1.24–9.45) | 3.05 (1.07–8.68) | 0.80 (0.19–3.40) | 0.60 (0.13–2.73) |

[a] Bacterial co-infection occurred in a total of 5 patients, all of which were bacteremia due to *E. faecalis* (n = 2), *K. aerogenes* (n = 1), *K. pneumoniae* (n = 1), and *P. aeruginosa* (n = 1).

[b] Invasive devices included central vascular catheter (n = 46), gastrostomy (n = 5), tracheostomy (n = 5), ventriculo-peritoneal shunt (n = 2).

Compared to the group with vancomycin MIC ≤ 1.0 µg/mL, persistent bacteremia at 48–72 hours and recurrent MRSA bacteremia were more frequently observed in those with vancomycin MIC 2.0 µg/mL although there was no statistical significance (33.3% vs 60.0%; *P* = 0.105, and 16.1% vs 40.0%; *P* = 0.077, respectively).

Compared to the initial vancomycin dose ≥ 60 mg/kg/day group, the <60 mg/kg/day group tended to have more cases of persistent bacteremia at 48–72 hours, but there was no statistical significance (17.6% vs 42.9%; *P* = 0.059). Clinical outcomes, including recurrent MRSA infection and 30-day all-cause mortality, were not influenced by the initial vancomycin dose.

## Risk factors for persistence of MRSA bacteremia and 30-day all-cause mortality

Unadjusted logistic regression analysis revealed initial $C_{trough}$ < 10 µg/mL and initial AUC/MIC < 300 were risk factors associated with persistent bacteremia at 48–72 hours [odds ratio (OR), 3.37; 95% confidence interval (CI), 1.00–11.41, and 3.43; 95% CI, 1.24–9.45, respectively] (Table 4).

Although initial $C_{trough}$ < 10 µg/mL tended to increase the risk of persistent bacteremia at 48–72 hours (adjusted OR, 3.14; 95% CI, 0.86–11.41), in multivariate analysis adjusting for age, co-infection, indwelling medical device, and presence of primary focus, initial AUC/MIC < 300 was the only statistically significant risk factor associated with persistent bacteremia at 48–72 hours (adjusted OR, 3.05; 95% CI, 1.07–8.68). Co-infection, indwelling medical device, ICU stay, concurrent antibiotics use, and vancomycin MIC were not statistically significant risk factors for persistent bacteremia in unadjusted and adjusted logistic regression analysis.

Furthermore, 30-day all-cause mortality was not influenced by initial $C_{trough}$ nor initial AUC/MIC by adjusted or unadjusted logistic regression analysis. Age, co-infection, indwelling medical device, and presence of primary focus of bacteremia were not risk factors associated with 30-day all-cause mortality under adjusted nor unadjusted logistic regression analysis.

## Discussion

In this study, we analyzed the relationship between initial vancomycin $C_{trough}$ and AUC/MIC, and the impact of these PK/PD parameters on clinical/microbiological outcomes of MRSA bacteremia in a pediatric population. Initial $C_{trough}$ < 10 µg/mL and AUC/MIC < 300 were

associated with microbiological failure at 48–72 hours of vancomycin use, and initial AUC/MIC < 300 nearly tripled the risk for persistent MRSA bacteremia in children. However, these initial PK/PD parameters were not correlated with clinical outcomes in terms of recurrent MRSA infection and all-cause mortality within 30 days.

Our previous study analyzed the impact of initial $C_{trough}$ on the clinical and microbiological outcomes in pediatric MRSA bacteremia without data regarding AUC/MIC [12]. Herein, we extended the previous study, including more cases with MRSA bacteremia and estimated the AUC/MIC using vancomycin dose, vancomycin clearance, patient's weight, and serum creatinine in a calculation proposed by Le et al. [5]. This study revealed a weak linear relationship between $C_{trough}$ and estimated AUC/MIC, but it is still inconclusive as to whether there is a positive correlation between $C_{trough}$ and AUC [16, 17]. A recent pediatric study suggested a positive correlation between $C_{trough}$ and AUC; a $C_{trough}$ of 10 μg/mL, 15 μg/mL, and 20 μg/mL can predict AUC of 413, 548, and 714, respectively [18]. However, Chhim et al. reported that the correlation between $C_{trough}$ and estimated AUC/MIC was poor ($R^2 = 0.14$–$0.20$) [19]. Among the various methods for calculating AUC, Le's equation tended to present the highest AUC value [17]. Extrapolation of AUC from serum $C_{trough}$ might underestimate AUC by up to 25%, and the AUC value varied between patients with similar results by up to 30-fold [8]. Accordingly, simultaneous monitoring of $C_{trough}$ and $C_{peak}$ has been suggested for accurate calculation of AUC/MIC. In a comparison of methods for calculating AUC, the trapezoidal method using both $C_{trough}$ and $C_{peak}$ could estimate AUC/MIC more accurately than the other methods [5, 16, 17]. Considering the trough-only concentration monitoring method might not reflect the actual AUC value in clinical settings and it is not always feasible to draw blood from young children at least two times per day, the Bayesian computer software programs could be used to estimate the appropriate vancomycin AUC value with minimal sampling [14, 20].

Although a recent study in adult patients showed that AUC/MIC of > 300 could be used as a surrogate marker for favorable clinical outcomes [10], a value of AUC/MIC 300–400 might be subtherapeutic, because AUC/MIC > 400 has been used as a standard surrogate marker for predicting the effectiveness of vancomycin use [21]. In this study, initial vancomycin AUC/MIC < 300 could be used as a predictor of persistent MRSA bacteremia at 48–72 hours, and AUC/MIC was a more reliable pharmacokinetic parameter predicting favorable microbiological outcomes compared to $C_{trough}$. However, given that the initial AUC/MIC > 400 was only achieved in 33% in this study, the role of MIC should be addressed in correlating AUC with MIC, dose, $C_{trough}$, and bacteremia clearance.

There is a controversy about a limited activity of vancomycin against MRSA isolates with a MIC at the high end of the susceptibility range [22, 23]; vancomycin MIC ≥ 1.5 μg/mL is associated with treatment failure in MRSA bacteremia [24]. In other study, however, the clinical outcomes of MRSA bacteremia were not different between vancomycin MIC < 2.0 μg/mL or 2.0 μg/mL [25]. Our study suggested that persistent or recurrent bacteremia tended to occur more often in the group with vancomycin MIC 2.0 μg/mL compared to those with MIC ≤ 1.0 μg/mL although there was no statistical significance. Even though it is difficult to say clearly that vancomycin "MIC creep" was observed during this study, which was defined as a gradual increase in the central tendency of the vancomycin MIC above 1 or 1.5 μg/mL for the dominant wild-type population [26], there was no increased trend of the proportion of MRSA isolates with vancomycin MIC > 1.0 μg/mL. However, it is important to clearly distinguish MIC creep from an increased occurrence of specified epidemic clones for which vancomycin MIC are elevated. So, it is necessary to monitor long-term changes in vancomycin MIC and the increase of specific clones with higher MIC. Additionally, there is considerable variability in MIC results between the susceptibility testing methods [27, 28], and the current

critical PK/PD index was accepted with the MIC determined by BMD [14]. Because our institution perform MIC testing using automated systems, MicroScan WalkAway, it may be helpful to consider performing BMD in future studies to minimize method-dependent differences in MIC results.

Recent studies of pediatric MRSA bacteremia suggested that a higher $C_{trough}$ does not correlate with clinical outcomes [11], and the therapeutic discordance between AUC and $C_{trough}$ may lead to suboptimal outcomes among patients with infections due to pathogens with higher MIC values [18, 20]. In addition, targeting vancomycin trough levels > 15 μg/mL in pediatrics could expose them to adverse effects such as nephrotoxicity or ototoxicity [18]. It is difficult to give clear meaning due to the small number, but in a total of 3 patients who experienced transient AKI in this study, the initial $C_{trough}$ were all < 10 μg/mL.

The recurrence rate was relatively high in this study, although it was not possible to clearly distinguish between recurrence or reinfection of MRSA bacteremia because genotyping analysis using pulsed field gel electrophoresis or multilocus sequence typing was not performed. This might be explained by a high proportion of patients with severe underlying disease and long-term hospitalization, and clinician's preference for salvage therapy for catheter associated infection.

This study had some limitations. Because of the small study population from a single center, its generalizability may be poor, and the low event rate of clinical outcome including mortality makes multivariable analysis challenging. We conducted this study retrospectively, and checked only $C_{trough}$ during vancomycin treatment as the standard of care. So, the estimated AUC without $C_{peak}$ might not reflect the actual AUC in pediatric patients. However, there have been few clinical studies to determine the impact of the PK/PD parameters of vancomycin on clinical and microbiological outcomes in MRSA bacteremia especially among children population. We adopted methods for calculating AUCs that could be applicable for children under 2 years old, and determined the impact of the vancomycin PK/PD parameters of $C_{trough}$, AUC, and MIC on clinical and microbiological outcomes in MRSA bacteremia which occurred in young children. Finally, this is a study of initial $C_{trough}$ or AUC/MIC, rather than a study of PK/PD parameters over time during a treatment course. Given the clinical significance of subsequent $C_{trough}$ or AUC/MIC, a delicate analysis of the relationship between PK/PD parameters and clinical/microbiological outcomes throughout the treatment period will be essential.

In conclusion, initial vancomycin AUC/MIC < 300 increased the risk for early microbiologic failure during treatment for pediatric MRSA bacteremia, although it did not impact 30-day fatalities or recurrence. In addition, $C_{trough}$ and AUC/MIC had only weak relationships in children. Future studies using more accurate PK/PD parameters are needed to determine the optimal vancomycin exposure and to improve the clinical and microbiological outcomes in children.

## Supporting information

**S1 Data.**
(XLSX)

## Acknowledgments

This manuscript was presented in part at the ASM microbe 2019, in San Francisco, California, USA, June 20–24, 2019.

## Author Contributions

**Conceptualization:** Reenar Yoo, Jina Lee.

**Data curation:** Reenar Yoo, Hyejin So, Euri Seo, Jina Lee.

**Formal analysis:** Reenar Yoo, Jina Lee.

**Investigation:** Reenar Yoo, Jina Lee.

**Methodology:** Euri Seo, Mina Kim.

**Writing – original draft:** Reenar Yoo, Jina Lee.

**Writing – review & editing:** Reenar Yoo, Jina Lee.

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
