## [Decision Letter · Decision Letter 0]

8 Oct 2020

PONE-D-20-22107

Impact of initial vancomycin pharmacokinetic/pharmacodynamic parameters on the clinical and microbiological outcomes of Methicillin-resistant *Staphylococcus aureus*bacteremia in children

PLOS ONE

Dear Dr. Lee,

Thank you for submitting your manuscript to PLOS ONE. After careful consideration, we feel that it has merit but does not fully meet PLOS ONE’s publication criteria as it currently stands. Therefore, we invite you to submit a revised version of the manuscript that addresses the points raised during the review process.

As you can see, both reviewers were positive about your study and made many constructive comments and recommendations. Please revise your manuscript by addressing their comments and suggestions.

We look forward to receiving your revised manuscript.

Kind regards,

Taeok Bae

Academic Editor

PLOS ONE

Journal Requirements:

2. Thank you for providing the following ethics information in the methods section of your manuscript “This study was approved by the Institutional Review Board of Asan Medical Center with a waiver of informed consent due to the study being retrospective and using de-identified data collection and analysis (IRB No. 2019-1638)”. Please also provide this information in the ethics statement on the online submission form

3.Thank you for stating the following financial disclosure:

 [The funders had no role in study design, data collection and analysis, decision to publish, or preparation of the manuscript.].

Reviewers' comments:

Reviewer's Responses to Questions

**Comments to the Author**

1. Is the manuscript technically sound, and do the data support the conclusions?

Reviewer #1: Yes

Reviewer #2: Yes

2. Has the statistical analysis been performed appropriately and rigorously? 

Reviewer #1: Yes

Reviewer #2: Yes

3. Have the authors made all data underlying the findings in their manuscript fully available?

Reviewer #1: Yes

Reviewer #2: Yes

4. Is the manuscript presented in an intelligible fashion and written in standard English?

Reviewer #1: Yes

Reviewer #2: Yes

5. Review Comments to the Author

Reviewer #1: Thank you for the opportunity to your manuscript.

Line 48-49 - Reword to "Vancomycin is an important treatment option...", as saying it's "an important treatment of choice" implies there are multiple choices.

Line 52 - Vancomycin is concentration dependent, not independent.

Line 55 - This is true when MIC is 1 or less than 1

Line 61 - Ctrough is correlated if MIC is 1 or less but otherwise it is correlated. Trough of 15-20 will achieve AUC/MIC > 400 for S. aureus > 90% of the time if MIC is 1 or less. The problem is the "MIC creep" in which AUC/MIC > 400 is more difficult to achieve. I suggest reviewing Pai et al. Advanced Drug Delivery Reviews. 2014;77:55-57 and Alsutan et al. Peditr Infect Dis J. 2018;37:880-885.

Line 106 - Why did you not define renal toxicity as also a need to adjust the vancomycin dose based on SCr? Or even a clinical definition of AKI? Would have been more useful to capture a more inclusive definition/data.

Line 125 - Did you calculate power? It is unclear to me which outcome is your primary outcome.

Line 176 - The phenomenon of "MIC creep" could have skewed your data. There is now a higher proportion of "susceptible" S. aureus isolates that have an MIC of 2 versus 0.5 or 1 and increased clinical failure.

Line 179 - You mention the creep here; how are you defining it?

213 - Make sure you have %s in the parentheses where appropriate throughout the entire chart.

Line 222 - Why did those whose vancomycin MIC was 1 or less have more persistent bacteremia? What confounders were at play? Dose differences?

Line 226 - This is an important finding. I would recommend also stratifying by > 80 mg/kg, as doses this high would warrant a switch to a different antibiotic. I would also stratify AUC > 400 attainment by < 60, 60-80, and > 80 mg/kg/day.

Line 273 - You reference a study showing AUC/MIC > 300 can be used as a marker of clinical efficacy, but the standard is > 400. A value of 300-400 would be subtherapeutic and even though you found an initial AUC/MIC > 300 is superior to < 300, you did not (as far as I can tell) address the role of the MIC. What is important is correlating AUC with MIC, dose, trough, and bacteremia clearance.

Line 293 - Why would you have to draw and peak and trough 2 hours apart for this method?

Line 297 - You mention there was no difference between MC Line 300 - You state there is no MIC creep; how are you defining this? 60% of patients with MIC 2 had persistent bacteremia. I would say that the fact that initial AUC > 400 was only achieved 33% of time could be due to MIC creep.

It is a privilege to review this manuscript. Thank you.

Reviewer #2: Yoo and colleagues have evaluated vancomycin PK/PD parameters in the treatment of pediatric MRSA bacteremia with regards to clinical and microbiological outcomes. In a retrospective single center cohort study over 8 years, they found that there was a weak relationship between trough concentration and AUC/MIC, and that initial AUC/MIC <300 was associated with persistent MRSA bacteremia at 48-72 hours. Despite the long study period, there were only 73 children in the final analysis and even fewer children who experienced the outcomes of interest. Given the dearth in PK/PD and clinical outcome data in pediatric S. aureus bacteremia, the authors are to be commended for their research, and show that the published AUC/MIC target of 400 for adults is also reasonable for children.

Comments:

Lines 72-73:

The authors proposed to study Ctrough and AUC/MIC for optimal dosing to achieve favorable clinical and microbiological outcomes. However this really is a study of initial Ctrough or AUC/MIC, rather than a study of PK/PD parameters over time during a treatment course. It maybe that subsequent Ctrough or AUC/MIC are also important in outcomes, but this has not been evaluated. This should be highlighted in the introduction and in the discussion.

Line 96:

Vancomycin MIC was performed using MicroScan. The establishment of AUC/MIC targets were established using BMD, and there are known method-based differences that lead to differences in MIC results. Was there any consideration for performing BMD?

Lines 108-109:

The definition of renal toxicity included … “the need to replace vancomycin with an alternate antibiotic because of serum creatinine elevation”. Although no patients experienced renal toxicity, the latter definition is very vague. How much serum creatinine elevation was required to meet this definition? Did it differ between treating clinicians? This may underestimate the true incidence of renal toxicity.

Line 114:

One of the clinical outcomes was “recurrence of MRSA bacteremia”. The definition for this outcome does not appear until the footnote of Table 2. Suggest insert into the Methods section here.

Lines 142-148, 164-170, 171-172:

These paragraphs are simply repeating information that is found in Table 1. There is no need to repeat data in the text that are also found in the Table. Suggest re-work text/table so that there is no repetition of material.

Lines 187-191:

The authors have reported mean Ctrough and AUC/MIC values. Were these normally distributed? Looking at the mean and range distributions provided, I suspect the data are non-parametric, so median values might be better.

Also, it is not clear that mean +/- SD are reported because SD has not been indicated in the text.

Lines 271-280:

The authors have referenced predominantly adult literature for PK/PD targets. The updated vancomycin consensus guideline has been released (Rybak Am J Health Syst Pharm 2020) and now includes pediatric sections. Suggest that this is included in discussion and references, rather than the superceded 2009 version. This updated guideline also recommends similar AUC/MIC targets to the adult population.

Table 1:

Please also include number of patients with 30-day mortality, not just percentage (to be consistent with formatting of the rest of the table). What was the rate of recurrent bacteremia?

Table 2:

Although it is not summarized elsewhere, the number of children with recurrent bacteremia was 14. From the denominator population of 62 children (presumably there were incomplete/missing data for the remaining 11 children), this is 23%. Additionally, there were 27 children with persistent bacteremia (37%). These are extraordinarily high rates of recurrent and persistent bacteremia. Do the authors have any explanation for these high rates?

Table 3:

Presentation of the adjusted multivariable model for persistent MRSAB and mortality should only include the final parsimonious model (ie. only variables with p<0.05).

Note that the small numbers of children with these outcomes (27 for persistent SAB and 9 for mortality) makes multivariable analysis challenging due to the low event rate. This should be added to the discussion.

Minor errors:

Line 126: Fischer should be spelt Fisher

Table 2:

The numbers in each group of AUC/MIC category should also be presented (as has been done for Ctrough)

The numbers in each group (persistent MRSAB, recurrent MRSAB, mortality) should be presented in the top columns.

Figure 1:

Both axes should have a unit of measurement presented.

6. PLOS authors have the option to publish the peer review history of their article (what does this mean?). If published, this will include your full peer review and any attached files.

Reviewer #1: No

Reviewer #2: No

---

## [Author Response · Author response to Decision Letter 0]

10 Jan 2021

Dear Taeok Bae, Academic Editor

Thank you for giving us a positive review of our study. We did our best to make corrections based on the opinions of the reviewers. 

Journal Requirements:

2. Thank you for providing the following ethics information in the methods section of your manuscript “This study was approved by the Institutional Review Board of Asan Medical Center with a waiver of informed consent due to the study being retrospective and using de-identified data collection and analysis (IRB No. 2019-1638)”. Please also provide this information in the ethics statement on the online submission form.

 => I wrote down the information in the ethics statement on the online submission form. Thanks. 

3.Thank you for stating the following financial disclosure:

 [The funders had no role in study design, data collection and analysis, decision to publish, or preparation of the manuscript.].

a. Please clarify the sources of funding (financial or material support) for your study. List the grants or organizations that supported your study, including funding received from your institution.

d. If you did not receive any funding for this study, please state: “The authors received no specific funding for this work.”

We received no specific funding for this work. We include this statement, “The authors received no specific funding for this work” within my cover letter. Thanks.

Reviewers' comments:

Reviewer's Responses to Questions

Comments to the Author

1. Is the manuscript technically sound, and do the data support the conclusions?

Reviewer #1: Yes

Reviewer #2: Yes

2. Has the statistical analysis been performed appropriately and rigorously? 

Reviewer #1: Yes

Reviewer #2: Yes

3. Have the authors made all data underlying the findings in their manuscript fully available?

Reviewer #1: Yes

Reviewer #2: Yes

4. Is the manuscript presented in an intelligible fashion and written in standard English?

Reviewer #1: Yes

Reviewer #2: Yes

5. Review Comments to the Author

Reviewer #1: Thank you for the opportunity to your manuscript.

Thank you very much for your positive assessment of our research and for all the informative advice.

Line 48-49 - Reword to "Vancomycin is an important treatment option...", as saying it's "an important treatment of choice" implies there are multiple choices.

I changed the sentence as you recommended. 

Line 50-51, “Vancomycin is an important treatment option for invasive MRSA infections.”

Line 52 - Vancomycin is concentration dependent, not independent.

Thank you for your comment. Raybak, et al. mentioned in a review paper in 2006 as follows; “Vancomycin is concentration-independent antibiotic” (Clin Infect Dis 2006;42:S35-S39). Larsson et al. also suggested that the administration of stepwise increasing clinical concentrations (range, 5–40 mg/L) resulted in no appreciable difference in killing even though the postantibiotic effect of vancomycin is dependent on the concentration (J Antimicrob Chemother. 1996;38:589-97). However, as you said, the serum concentration of vancomycin is used as one of the important indicators of vancomycin PK/PD. Considering this sentence may be confusing to readers, I deleted the sentence of “vancomycin is a concentration-independent antibiotic” in the revised manuscript.

Line 55 - This is true when MIC is 1 or less than 1

Thank you for your comment. I corrected the typo as follows; 

Line 56-57, “Although there is an assumption that a vancomycin C trough of > 15 µg/mL when the MIC is ≤ 1 µg/mL,”

Line 61 - Ctrough is correlated if MIC is 1 or less but otherwise it is correlated. Trough of 15-20 will achieve AUC/MIC > 400 for S. aureus > 90% of the time if MIC is 1 or less. The problem is the "MIC creep" in which AUC/MIC > 400 is more difficult to achieve. I suggest reviewing Pai et al. Advanced Drug Delivery Reviews. 2014;77:55-57 and Alsutan et al. Peditr Infect Dis J. 2018;37:880-885.

Thank you for sharing important points and references. As mentioned, this situation is only applicable to MRSA with MIC ≤ 1 µg/mL which is determined by broth microdilution (BMD). Therefore, it is stated that it is meaningful only in the case of MIC ≤ 1 µg/mL as follow; 

Line 56-58, vancomycin C trough of > 15 µg/mL when the MIC is ≤ 1 µg/mL can be used as a good surrogate marker for AUC/MIC ratio of ≥ 400 with the MIC determined by broth microdilution (BMD) for achieving clinical effectiveness in adults [4]

Line 61-63, “vancomycin C trough 7–10 μg/mL could predict the achievement of AUC/MIC > 400 at a dose of 15 mg/kg/dose every 6 hours in the case of patients with normal renal function, if the MIC ≤ 1 µg/mL [6]”

As with the opinion of another reviewer, the ASHP report released in 2020 has also been added as a major reference along with the two papers above.

Line 106 - Why did you not define renal toxicity as also a need to adjust the vancomycin dose based on SCr? Or even a clinical definition of AKI? Would have been more useful to capture a more inclusive definition/data.

Thank you for comments. Most commonly used definition of vancomycin associated AKI is an increase in the serum creatinine level of 0.5mg/dL, or a 50% increase from baseline in consecutive daily readings, or a decreased in calculated creatinine CL(CLcr) of 50% from baseline on 2 consecutive days in the absence of a alternative explanation (Raybak, et al. ASHP report 2020). 

Since this study included young infants, the initial Cr value may be low, for example, if the serum Cr is doubled to 0.2 mg/dL from 0.1 mg/dL, it is too much to consider as AKI. According to your comments, the definition of AKI is also added in the manuscript as follows: “AKI was defined by an increase in the serum creatinine level of 0.5 mg/dL.” If AKI was defined as a rise of 0.5 mg/dL or more in serum Cr, AKI occurred in 3 out of 73 people in this study. In addition, the previously mentioned “renal toxicity” was modified to a “clinically significant nephrotoxicity”.

Line 113-117, Acute kidney injury (AKI) was defined as an increase in serum creatinine levels by 0.5 mg/dL (Raybak, et al. ASHP report 2020). The clinically significant renal toxicity of vancomycin was defined as an additional need for renal replacement therapy during vancomycin treatment or the need to replace vancomycin with an alternate antibiotic because of serum creatinine elevation. 

Line 125 - Did you calculate power? It is unclear to me which outcome is your primary outcome.

Thank you for comments. This study was not based on data collected prospectively, but was a retrospective study of pediatric MRSA bacteremia during the last 8 years, when vancomycin TDM was actively performed. Therefore, statistical power was not accurately calculated in advance.

According to your comments, I clarified the primary outcome as follows: 

Line 100-101, The primary outcome of this study is to clarify the relationship between the initial C trough and AUC/MIC, and the potential impact of the variables on clinical and microbiological outcomes.

Line 176 - The phenomenon of "MIC creep" could have skewed your data. There is now a higher proportion of "susceptible" S. aureus isolates that have an MIC of 2 versus 0.5 or 1 and increased clinical failure.

I agree with your opinion. In this study, the persistent and recurrent bacteremia were observed more frequently in the group with vancomycin MIC 2.0 μg/mL although there was no statistical significance. With the MIC being a component of the vancomycin AUC/MIC targeted surrogate for efficacy, it is important to be aware of vancomycin susceptibility patterns for MRSA. In our institute, the proportion of S. aureus isolates with MIC 1 μg/mL or greater than 1 μg/mL, which was determined by MicroScan WalkAway showed an increasing trend. Table 2, which shows the annual trend of the vancomycin MIC of MRSA isolates, is also added in the revised manuscript. 

Line 179 - You mention the creep here; how are you defining it?

Thanks for pointing out. I have used inappropriate terms in the previous manuscript because vancomycin MIC “creep” can be defined as a gradual increase in the central tendency of the vancomycin MIC for the dominant wild-type population (Sader, et al. AAC 2009). Furthermore, an increase in the proportion of strains for which the vancomycin MIC is >1 or >1.5 ug/mL may be caused by either extensive use of vancomycin or more likely when dealing with S. aureus, the dissemination of a clone or clones with less susceptibility to vancomycin. Thus, it is important to clearly distinguish MIC creep from an increased occurrence of specified epidemic clones for which vancomycin MIC are elevated.

The following sentences in the previous manuscript were deleted to minimize confusion.

Line 188, “Although no definite creep of vancomycin MIC was observed” 

 Line 349-350 “In addition, there was no significant vancomycin MIC creep phenomenon during this study period”

Instead of the term “MIC creep”, the following sentences were added in the revised manuscript.

Line 193-195, During the study period, the proportion of MRSA isolates with vancomycin MIC � 0.5 μg/mL showed a decreasing trend; 30.8% (2010-2012), 3.3% (2013–2015), and 5.9% (2016–2018) (P for trend =0.246).

Line 342-350 Our study suggested that persistent or recurrent bacteremia tended to occur more often in the group with vancomycin MIC 2.0 μg/mL compared to those with MIC ≤ 1.0 μg/mL although there was no statistical significance. Even though it is difficult to say clearly that vancomycin MIC creep was observed during this study, the proportion of MRSA isolates with vancomycin MIC 1.0 μg/mL or greater than 1.0 μg/mL showed an increasing trend. However, it is important to clearly distinguish MIC creep from an increased occurrence of specified epidemic clones for which vancomycin MIC are elevated [26]. So, it is necessary to monitor long-term changes in vancomycin MIC and the increase of specific clones with higher MIC.

213 - Make sure you have %s in the parentheses where appropriate throughout the entire chart.

I checked the numbers and %s in tables. 

Line 222 - Why did those whose vancomycin MIC was 1 or less have more persistent bacteremia? What confounders were at play? Dose differences?

Sorry for the confusion. In the revised manuscript, it was revised and described as follow:

Line 267-270 Compared to the group with vancomycin MIC ≤ 1.0 μg/mL, persistent bacteremia at 48–72 hours and recurrent MRSA bacteremia were more frequently observed in those with vancomycin MIC 2.0 μg/mL although there was no statistical significance (33.3% vs 60.0%; P=0.105, and 16.1% vs 40.0%; P=0.077, respectively). 

In addition, I corrected errors of numbers in table 3 in the revised manuscript. 

Line 226 - This is an important finding. I would recommend also stratifying by > 80 mg/kg, as doses this high would warrant a switch to a different antibiotic. I would also stratify AUC > 400 attainment by < 60, 60-80, and > 80 mg/kg/day.

In this study, only the initial dose of vancomycin was analyzed, and no initial dose was given more than 80mg/kg a day. So, no further analysis could proceed.

Line 273 - You reference a study showing AUC/MIC > 300 can be used as a marker of clinical efficacy, but the standard is > 400. A value of 300-400 would be subtherapeutic and even though you found an initial AUC/MIC > 300 is superior to < 300, you did not (as far as I can tell) address the role of the MIC. What is important is correlating AUC with MIC, dose, trough, and bacteremia clearance.

Thank you for mentioning a good point. According to your advice, I modified it as follows: many parts in the discussion 

Line 329- 337, Although a recent study in adult patients showed that AUC/MIC of > 300 could be used as a surrogate marker for favorable clinical outcomes [10], a value of AUC/MIC 300-400 might be subtherapeutic, because AUC/MIC > 400 has been used as a standard surrogate marker for predicting the effectiveness of vancomycin use [18]. In this study, initial vancomycin AUC/MIC < 300 could be used as a predictor of persistent MRSA bacteremia at 48–72 hrs, and AUC/MIC was a more reliable pharmacokinetic parameter predicting favorable microbiological outcomes compared to C trough. However, given that the initial AUC/MIC > 400 was only achieved in 33% in this study, the role of MIC should be addressed in correlating AUC with MIC, dose, C trough, and bacteremia clearance.

Line 293 - Why would you have to draw and peak and trough 2 hours apart for this method?

Sorry for confusion. I modified the previous sentence as follows:

Line 326, it is not always feasible to draw blood from young children at least two times per day

Line 297 - You mention there was no difference between MC Line 300 - You state there is no MIC creep; how are you defining this? 60% of patients with MIC 2 had persistent bacteremia. I would say that the fact that initial AUC > 400 was only achieved 33% of time could be due to MIC creep.

Thanks a lot for a good point. According to your advice, I modified lots of part in the discussion in the revised manuscript as follows. 

Line 329-344, Although a recent study in adult patients showed that AUC/MIC of > 300 could be used as a surrogate marker for favorable clinical outcomes [10], a value of AUC/MIC 300-400 might be subtherapeutic, because AUC/MIC > 400 has been used as a standard surrogate marker for predicting the effectiveness of vancomycin use [18]. In this study, initial vancomycin AUC/MIC < 300 could be used as a predictor of persistent MRSA bacteremia at 48–72 hrs, and AUC/MIC was a more reliable pharmacokinetic parameter predicting favorable microbiological outcomes compared to C trough. However, given that the initial AUC/MIC > 400 was only achieved in 33% in this study, the role of MIC should be addressed in correlating AUC with MIC, dose, C trough, and bacteremia clearance.

There is a controversy about a limited activity of vancomycin against MRSA isolates with a MIC at the high end of the susceptibility range [22, 23]; vancomycin MIC ≥ 1.5 μg/mL is associated with treatment failure in MRSA bacteremia [24]. In other study, however, the clinical outcomes of MRSA bacteremia were not different between vancomycin MIC < 2.0 μg/mL or 2.0 μg/mL [25]. Our study suggested that persistent or recurrent bacteremia tended to occur more often in the group with vancomycin MIC 2.0 μg/mL compared to those with MIC ≤ 1.0 μg/mL although there was no statistical significance.

In addition, the previous sentence of “Clinical and microbiologic outcomes were not influenced by vancomycin MIC.” was deleted in the revised manuscript. 

Reviewer #2: Yoo and colleagues have evaluated vancomycin PK/PD parameters in the treatment of pediatric MRSA bacteremia with regards to clinical and microbiological outcomes. In a retrospective single center cohort study over 8 years, they found that there was a weak relationship between trough concentration and AUC/MIC, and that initial AUC/MIC <300 was associated with persistent MRSA bacteremia at 48-72 hours. Despite the long study period, there were only 73 children in the final analysis and even fewer children who experienced the outcomes of interest. Given the dearth in PK/PD and clinical outcome data in pediatric S. aureus bacteremia, the authors are to be commended for their research, and show that the published AUC/MIC target of 400 for adults is also reasonable for children.

Thank you for your positive evaluation of our research. The revised version is supplemented by reflecting the informative opinions you mentioned. Thank you again. 

Comments:

Lines 72-73:

The authors proposed to study Ctrough and AUC/MIC for optimal dosing to achieve favorable clinical and microbiological outcomes. However this really is a study of initial Ctrough or AUC/MIC, rather than a study of PK/PD parameters over time during a treatment course. It maybe that subsequent Ctrough or AUC/MIC are also important in outcomes, but this has not been evaluated. This should be highlighted in the introduction and in the discussion.

Thanks for your comments. To avoid misinterpretation, the expression “initial” is marked in front of both the Ctrough and the AUC used in this study. In addition, the following sentences as you mentioned were added as one of the drawbacks in this study.

Line 378-382, Finally, this is a study of initial Ctrough or AUC/MIC, rather than a study of PK/PD parameters over time during a treatment course. Given the clinical significance of subsequent Ctrough or AUC/MIC, a delicate analysis of the relationship between PK/PD parameters and clinical/microbiological outcomes throughout the treatment period will be essential. 

Line 96:

Vancomycin MIC was performed using MicroScan. The establishment of AUC/MIC targets were established using BMD, and there are known method-based differences that lead to differences in MIC results. Was there any consideration for performing BMD?

Thank you for mentioning a good point. In considering the differences between these methods of MIC testing, I described the following: 

Line 350-354, Additionally, there is considerable variability in MIC results between the susceptibility testing methods [27, 28], and the current critical PK/PD index was accepted with the MIC determined by BMD [14]. Because our institution perform MIC testing using automated systems, MicroScan WalkAway, it may be helpful to consider performing BMD in future studies to minimize method-dependent differences in MIC results.

Although I could not analyze BMD results further in this study, I will try to conduct BMD analysis together in future studies. Thank you. 

Lines 108-109:

The definition of renal toxicity included … “the need to replace vancomycin with an alternate antibiotic because of serum creatinine elevation”. Although no patients experienced renal toxicity, the latter definition is very vague. How much serum creatinine elevation was required to meet this definition? Did it differ between treating clinicians? This may underestimate the true incidence of renal toxicity.

Thank you for the comments. It was also mentioned in the response to Reviewer 1`s advice, the following definitions of acute kidney injury were added

Line 113-117, Acute kidney injury (AKI) was defined as an increase in serum creatinine levels by 0.5 mg/dL [14]. The clinically significant renal toxicity of vancomycin was defined as an additional need for renal replacement therapy during vancomycin treatment or the need to replace vancomycin with an alternate antibiotic because of serum creatinine elevation. 

Line 185-187, Although AKI occurred in 3 out of 73 in this study, there was no clinically significant renal toxicity associated with vancomycin use. 

Line 114:

One of the clinical outcomes was “recurrence of MRSA bacteremia”. The definition for this outcome does not appear until the footnote of Table 2. Suggest insert into the Methods section here.

Thanks for the comments. The following was added to the related content method section. 

Line 108-110, Recurrent MRSA bacteremia was defined as another MRSA bacteremia event 1 month after MRSA bacteremia resolution.

Lines 142-148, 164-170, 171-172:

These paragraphs are simply repeating information that is found in Table 1. There is no need to repeat data in the text that are also found in the Table. Suggest re-work text/table so that there is no repetition of material.

Thanks for the comments. In order to reduce overlapping content and make it clearer, the following sentences were deleted in the revised manuscript.

In the previous manuscript (line 152-157) “including congenital heart disease (28.8%, 21 out of 73), malignancy (16.4%, 12 out of 73), multiple anomalies (6.8%, 5 out of 73), chronic lung disease (5.5%, 4 out of 73), neurologic disease (5.5%, 4 out of 73), and prematurity (4.1%, 3 out of 73). Most of the MRSA bacteremia cases (98.6%; 72 out of 73) were healthcare-associated infections and indwelling medical devices were involved in 79.5% (58 out of 73).”

Lines 187-191:

The authors have reported mean Ctrough and AUC/MIC values. Were these normally distributed? Looking at the mean and range distributions provided, I suspect the data are non-parametric, so median values might be better.

Also, it is not clear that mean +/- SD are reported because SD has not been indicated in the text.

Thank you for the comments. I modified to median value and interquartile range instead of mean+/- standard deviation in the revised manuscript. 

Lines 271-280:

The authors have referenced predominantly adult literature for PK/PD targets. The updated vancomycin consensus guideline has been released (Rybak Am J Health Syst Pharm 2020) and now includes pediatric sections. Suggest that this is included in discussion and references, rather than the superceded 2009 version. This updated guideline also recommends similar AUC/MIC targets to the adult population.

At the time of first submission of this paper, this guideline was not published. However, at the time of receiving this review opinion, this major guideline was released, so these contents were reflected and added to the reference.

Table 1:

Please also include number of patients with 30-day mortality, not just percentage (to be consistent with formatting of the rest of the table). What was the rate of recurrent bacteremia?

=> I corrected the numbers of fatal cases and denominators of clinical outcomes after reviewing the medical record again. One out of 73 patients with MRSA bacteremia was transferred to another hospital during treatment, allowing analysis of mortality and recurrence rates in a total of 72 patients: all-cause 30-day mortality was 9.7% (7 out of 72). Recurrent MRSA bacteremia occurred in 19.4% (14/72).

Table 2: (Table 3 in the revised manuscript)

Although it is not summarized elsewhere, the number of children with recurrent bacteremia was 14. From the denominator population of 62 children (presumably there were incomplete/missing data for the remaining 11 children), this is 23%. Additionally, there were 27 children with persistent bacteremia (37%). These are extraordinarily high rates of recurrent and persistent bacteremia. Do the authors have any explanation for these high rates?

Thank you for the comments. The denominators of recurrent bacteremia and fatality was modified clinical results in Table 3 in the revised manuscript. 

According to your advice and comments, it was described in the discussion section as follows;

Line 362-367, The recurrence rate was relatively high in this study, although it was not possible to clearly distinguish between recurrence or reinfection of MRSA bacteremia because genotyping analysis using pulsed field gel electrophoresis or multilocus sequence typing was not performed. This might be explained by a high proportion of patients with severe underlying disease and long-term hospitalization, and clinician`s preference for salvage therapy for catheter associated infection. 

Table 3: (Table 4 in the revised manuscript)

Presentation of the adjusted multivariable model for persistent MRSAB and mortality should only include the final parsimonious model (ie. only variables with p<0.05).

Note that the small numbers of children with these outcomes (27 for persistent SAB and 9 for mortality) makes multivariable analysis challenging due to the low event rate. This should be added to the discussion.

=> Thanks for your advice. According to your comments, I modified the table in the revised manuscprit. The variables included in this analysis are limited to age, bacaaterial co-infection, presence of invasive device, presence of primary focus, initial C trough, and initial AUC/MIC. Except for the variables of initial C trough and initial AUC/MIC, the other variables did not show statistical significance in the univariate analysis. 

The sentence you mentioned was added to the discussion. 

Line 369-370, the low event rate of clinical outcome including mortality makes multivariable analysis challenging.

Thanks again for all your comments. 

Minor errors:

Line 126: Fischer should be spelt Fisher

Thank you. I corrected the typo.

Table 2:

The numbers in each group of AUC/MIC category should also be presented (as has been done for Ctrough)

The numbers in each group (persistent MRSAB, recurrent MRSAB, mortality) should be presented in the top columns.

As you mentioned, I wrote down the numbers in each category. Thank you. 

Figure 1:

Both axes should have a unit of measurement presented.

Thanks. Units of measure are written on x-axis (ug/mL)

6. PLOS authors have the option to publish the peer review history of their article (what does this mean?). If published, this will include your full peer review and any attached files.

Do you want your identity to be public for this peer review? For information about this choice, including consent withdrawal, please see our Privacy Policy.

Reviewer #1: No

Reviewer #2: No

---

## [Decision Letter · Decision Letter 1]

27 Jan 2021

PONE-D-20-22107R1

Impact of Initial Vancomycin Pharmacokinetic/Pharmacodynamic Parameters on the Clinical and Microbiological Outcomes of Methicillin-Resistant Staphylococcus aureus Bacteremia in Children

PLOS ONE

Dear Dr. Lee,

Thank you for submitting your manuscript to PLOS ONE. After careful consideration, we feel that it has merit but does not fully meet PLOS ONE’s publication criteria as it currently stands. Therefore, we invite you to submit a revised version of the manuscript that addresses the points raised during the review process.

Since the raised points are mostly about formatting, your manuscript will proceed to the next step without further review if you address them sufficiently.

We look forward to receiving your revised manuscript.

Kind regards,

Taeok Bae

Academic Editor

PLOS ONE

Reviewers' comments:

Reviewer's Responses to Questions

**Comments to the Author**

1. If the authors have adequately addressed your comments raised in a previous round of review and you feel that this manuscript is now acceptable for publication, you may indicate that here to bypass the “Comments to the Author” section, enter your conflict of interest statement in the “Confidential to Editor” section, and submit your "Accept" recommendation.

Reviewer #1: All comments have been addressed

Reviewer #2: All comments have been addressed

2. Is the manuscript technically sound, and do the data support the conclusions?

Reviewer #1: Yes

Reviewer #2: Yes

3. Has the statistical analysis been performed appropriately and rigorously? 

Reviewer #1: Yes

Reviewer #2: Yes

4. Have the authors made all data underlying the findings in their manuscript fully available?

Reviewer #1: Yes

Reviewer #2: Yes

5. Is the manuscript presented in an intelligible fashion and written in standard English?

Reviewer #1: Yes

Reviewer #2: Yes

6. Review Comments to the Author

Reviewer #1: Thank you for addressing my comments. It is a pleasure to review your updated manuscript.

1. Abstract, line 32-33: Rephrase to make clear C trough was measured X times within the time period of Jan. 2010 to March 2018.

2. Abstract, line 36: Make clear that the 47.5% you are referring to is for mechanically ventilated patients (vs ICU admissions). Adding a comma after "intensive care unit stay" should help to clarify that.

3. Abstract, line 37 and 45: "mortality" used more commonly than "fatality"

4. Abstract, line 41, 44, 47: In general, I am noticing "hours" is abbreviated "hrs" or "hr". This should be consistent throughout; I would probably just keep the word spelled out as "hours" in your manuscript text.

5. Introduction, line 51: "major health concerns" is very broad/indirect. One of the major health concerns for what? Pediatrics? Hospitalized patients?

6. Introduction, line 56: cite

7. Introduction, line 58: You say "studies" but only give one citation

8. Discussion, page 20, line 4: "good relationship" should be changed to "positive correlation" or something else more objective.

9. Discussion, page 20, line 5: "good linear relationship" - same comment as above

10. Discussion, page 21, line 10: Please use quotations and define "MIC creep" when first introduced in your manuscript.

11. Discussion, page 21, line 11: Use >/= consistently vs switching to spelling out "greater than". Also, "MIC creep" is when MIC is > 1, not equal to or greater than 1, and still reported as susceptible.

12. Discussion, page 21, line 23-24: Rephrase; "could expose them to adverse effects such as nephrotoxicity or ototoxicity" or something similar that implies the potential only (and ototoxicity should be mentioned)

Thank you for addressing my previous comments so thoroughly. I am grateful for the opportunity to review this manuscript again.

Reviewer #2: I am satisfied that the authors have made revisions according to previous reviewer comments submitted.

7. PLOS authors have the option to publish the peer review history of their article (what does this mean?). If published, this will include your full peer review and any attached files.

Reviewer #1: No

Reviewer #2: No

---

## [Author Response · Author response to Decision Letter 1]

10 Feb 2021

Reviewer #1: Thank you for addressing my comments. It is a pleasure to review your updated manuscript.

1. Abstract, line 32-33: Rephrase to make clear C trough was measured X times within the time period of Jan. 2010 to March 2018.

Thank you for the comments. Only 89% (65/73) of the patients measured Ctrough at least twice. To clarify, I modified the sentence as follows; 

C trough was measured at least one time within the time period of January 2010 to March 2018.

2. Abstract, line 36: Make clear that the 47.5% you are referring to is for mechanically ventilated patients (vs ICU admissions). Adding a comma after "intensive care unit stay" should help to clarify that.

To make it clear, I changed the sentence as follows; 

Previously, “Severe cases required intensive care unit stay and mechanical ventilation occurred in 47.5% (35/73)

In this revised one, “Severe clinical outcomes leading to intensive care unit stay and/or use of mechanical ventilation occurred in 47.5% (35/73)

3. Abstract, line 37 and 45: "mortality" used more commonly than "fatality"

=> Thanks for the comments. Throughout the text, we have modified the term fatality by substituting the term mortality. 

4. Abstract, line 41, 44, 47: In general, I am noticing "hours" is abbreviated "hrs" or "hr". This should be consistent throughout; I would probably just keep the word spelled out as "hours" in your manuscript text.

According to the opinion, it was unified and revised in “hours”. Thanks. 

5. Introduction, line 51: "major health concerns" is very broad/indirect. One of the major health concerns for what? Pediatrics? Hospitalized patients?

=> Thanks for the comments. To make it clear, it has been modified as follows; 

In the previous manuscript, “Methicillin-resistant Staphylococcus aureus (MRSA) is one of the major health concerns,”

In the revised one, “Methicillin-resistant Staphylococcus aureus (MRSA) is one of the major health concerns in both healthcare-associated and community-acquired infections in children,”

6. Introduction, line 56: cite

Reference 1 was added after the sentence as follows; Vancomycin is an important treatment option for invasive MRSA infections [1].

7. Introduction, line 58: You say "studies" but only give one citation

=> Reference 3 is a review paper on PK/PD of vancomycin. Thanks. 

8. Discussion, page 20, line 4: "good relationship" should be changed to "positive correlation" or something else more objective.

I changed it to “positive correlation” instead of “good relationship”.

9. Discussion, page 20, line 5: "good linear relationship" - same comment as above

I changed it to “positive correlation” instead of “good linear relationship”

10. Discussion, page 21, line 10: Please use quotations and define "MIC creep" when first introduced in your manuscript.

According to your comments, I modified the previous sentences and use as follows; 

Even though it is difficult to say clearly that vancomycin “MIC creep” was observed during this study, which was defined as a gradual increase in the central tendency of the vancomycin MIC above 1 or 1.5 μg/mL for the dominant wild-type population [26],

11. Discussion, page 21, line 11: Use >/= consistently vs switching to spelling out "greater than". Also, "MIC creep" is when MIC is > 1, not equal to or greater than 1, and still reported as susceptible.

Thanks for the points. I modified the sentence as follows;

there was no increased trend of the proportion of MRSA isolates with vancomycin MIC > 1.0 μg/mL.

12. Discussion, page 21, line 23-24: Rephrase; "could expose them to adverse effects such as nephrotoxicity or ototoxicity" or something similar that implies the potential only (and ototoxicity should be mentioned)

=> Thank you for the comments. We changed the previous sentence as follows; 

In the previous manuscript, “In addition, targeting vancomycin trough levels > 15 µg/mL in pediatrics would expose them to unnecessary adverse events of nephrotoxicity.”

In the revised one, “In addition, targeting vancomycin trough levels > 15 µg/mL in pediatrics could expose them to adverse effects such as nephrotoxicity or ototoxicity.”

Thank you for addressing my previous comments so thoroughly. I am grateful for the opportunity to review this manuscript again.

We sincerely appreciate your meticulous review of our manuscript and great expert opinion. 

Reviewer #2: I am satisfied that the authors have made revisions according to previous reviewer comments submitted.

Thank you for reviewing our manuscript and for your informative expert opinion. Thank you again.

---

## [Editor Report · Decision Letter 2]

12 Feb 2021

Impact of Initial Vancomycin Pharmacokinetic/Pharmacodynamic Parameters on the Clinical and Microbiological Outcomes of Methicillin-Resistant Staphylococcus aureus Bacteremia in Children

PONE-D-20-22107R2

Dear Dr. Lee,

We’re pleased to inform you that your manuscript has been judged scientifically suitable for publication and will be formally accepted for publication once it meets all outstanding technical requirements.

Kind regards,

Taeok Bae

Academic Editor

PLOS ONE
---

## [Editor Report · Acceptance letter]

24 Mar 2021

PONE-D-20-22107R2 

Impact of Initial Vancomycin Pharmacokinetic/Pharmacodynamic Parameters on the Clinical and Microbiological Outcomes of Methicillin-Resistant *Staphylococcus aureus* Bacteremia in Children 

Dear Dr. Lee:

I'm pleased to inform you that your manuscript has been deemed suitable for publication in PLOS ONE. Congratulations! Your manuscript is now with our production department. 

Kind regards, 

on behalf of

Dr. Taeok Bae 

Academic Editor

PLOS ONE